# Transcription Factor and Zeatin Co-Regulate Mixed Catkin Differentiation of Chinese Chestnut (*Castanea mollissima*)

Xuan Zhou [1], Lu Wang [1], Qian Yin [1], Xinghui Liu [1], Joseph Masabni [2], Huan Xiong [1], Deyi Yuan [1] and Feng Zou [1,*]

[1] Key Laboratory of Cultivation and Protection for Non-Wood Forest Trees, Ministry of Education, Central South University of Forestry and Technology, Changsha 410004, China; 20211100075@csuft.edu.cn (X.Z.); leiry0318@outlook.com (L.W.); 20221100088@csuft.edu.cn (Q.Y.); 20221200119@csuft.edu.cn (X.L.); t20202524@csuft.edu.cn (H.X.); t20061123@csuft.edu.cn (D.Y.)

[2] Texas A & M Agri Life Research and Extension Center, 17360 Coit Rd, Dallas, TX 75252, USA; joe.masabni@ag.tamu.edu

* Correspondence: t20142217@csuft.edu.cn

**Abstract:** *Castanea mollissima* is an important monoecious fruit crop with high economic and ecological value in China. However, its yield is restricted by an imbalanced ratio of male and female flowers for chestnut production. To address this issue, we examined the morphology of bisexual flower organs, measured the levels of endogenous hormones in the flowers, profiled gene expression related to plant hormone biosynthesis and signaling pathways and transcription factors, and investigated the effects of exogenous jasmonic acid (JA) and zeatin (ZT) hormone application on flower development in *C. mollissima* 'Tanqiao'. Morphological studies indicated that the development of male and female flowers can be divided into nine and eight stages, respectively. Male flowers contained higher levels of gibberellic acid ($GA_3$) and abscisic acid (ABA) than female flowers, whereas female flowers had higher levels of JA and ZT. The analysis of the Kyoto encyclopedia of genes and genomes (KEGG) pathways revealed that the major significant enrichment pathways of differentially expressed genes (DEGs) consisted of plant hormone signal transduction and zeatin biosynthesis. Through time-series analyses, we screened 3 genes related to jasmonic acid biosynthesis and signal transduction and 21 genes related to zeatin biosynthesis and transduction. Among these genes, only the gene family *LOG*, related to zeatin biosynthesis, was highly expressed in female flowers. This result indicated that *LOG* may be the core gene hormone family involved in regulating female flower development. However, a weighted gene co-expression network analysis (WGCNA) suggested that *IDD7* was the core gene involved in regulating female flower development. The results of exogenous hormone application indicated that zeatin could greatly increase the quantity of fertile female flowers, but JA was not significant. These findings demonstrated that zeatin and transcription factors were crucial regulators in the formation of female flowers in *C. mollissima*.

**Keywords:** *Castanea mollissima*; floral morphology; phytohormone; RAN-seq; flower differentiation

## 1. Introduction

Flower differentiation is an essential period that directly impacts the number and quality of flowers, even influencing yield [1]. Hormones have a key function in flowering regulation [2]. The contents of abscisic acid (ABA), gibberellic acid ($GA_3$), and trans-Zeatin-riboside (ZR) in the buds of *Lycium ruthenicum* Murr. increased considerably during flower differentiation [3]. The levels of auxin (IAA) and ZR in female flowers are consistently higher than those in male flowers during flower differentiation in *Eucommia ulmoides* Oliv. [4]. Synergistic and antagonistic interactions have been observed among many hormones. Also, the induction of the flower differentiation in apple was connected to intricate hormone regulatory networks implicated in the cytokinin (CK), ABA, and $GA_3$ pathways [2]. Studies have shown that flower-differentiation signals may be mediated by hormone regulation [5–10]. For instance, the *JMSAUR* gene related to salicylic

acid was the core gene in regulating the sexual differentiation of flower buds in *Juglans mandshurica* [6]. In loquat, *TFL1*, *AP1*, and *FT* could respond to GA$_3$ signals to regulate flower differentiation [7]. In addition, exogenous phytohormones could affect flower differentiation. *MdGA20ox* was significantly inhibited by high levels of GA$_3$ in apple buds, reducing the flowering rate of apples [8]. The injection of the plant growth regulator Ethrel into a male sterile tree of *Diospyros kaki* Thunb. resulted in a feminizing effect [9]. If the level of the exogenous phytohormone reached a relatively high level, such as 640 mg/L of 6-Benzylaminopurine (6-BA), the phytohormone transformed the male inflorescences of *Vernicia fordii* into female inflorescences [10].

In addition, transcription factors (TFs) are also essential regulators associated with flower differentiation. In *Rafflesia cantleyi*, the expression of *MYB*, *WRKY*, *EriF*, and *bHLH* was expressed differentially and significantly during three different floral bud stages, at a higher level than flower differentiation [11]. Citrus flower differentiation was negatively regulated by HD-ZIP I Transcription Factor *PtHB13* when it was binding to the promoter of the flowering locus [12]. Recently, research has shown that TFs did not directly determine the beginning of flower differentiation in woody plants, instead determining the dynamic balance of various factors that regulated flower differentiation. A study showed that *MYC*, *FT*, *SOC1*, and *LFY* co-regulated flowering differentiation with endogenous hormones in *Camellia sinensis* [13]. Plant-hormone-related genes and some TFs co-expressed with *PpTFL1* in *Pyrus pyrifolia Nakai* were perhaps implicated in the *PpTFL1*-mediated floral induction [14]. GA$_3$ and ABA signals could inhibit flower differentiation by inversely regulating the expression of *MADS-box* gene in gloxinia [15]. In addition, the overexpression of *GHMYB24* in *Arabidopsis thaliana* could result in male sterility. The yeast two-hybrid showed that *GHMYB24* could interact with *GHJAZ1/2* to affect the jasmonic acid pathway and affect stamen development [16]. TFs and hormones may co-regulate flower-differentiation processes in plants.

*Castanea* is widely cultivated in the northern hemisphere in Asia, Europe, and Africa. *Castanea* is mainly planted in Hebei, Shandong, and Hubei provinces in China, which is one of the major producing countries, with a total planting area of over one million acres [17]. *Castanea* is rich in nutrients with edible nuts and timber value [18,19]. As a monoecious tree, the yield is not stable due to an imbalance in the ratio of male and female flowers. Thus, understanding the mechanism of flower differentiation was critical for enhancing the number of female flowers. Recently, Zhang et al. [20] identified five *FT/TFL1-like* genes in the chestnut genome and speculated that *FT* was the major gene involved in the morphogenesis of male and female flowers. This finding was verified by Cheng et al. [21], who observed that the overexpression of *CmFT* can promote flowering in *Arabidopsis*. In addition, phytohormones are also involved in *C. mollissima* flower differentiation. Cheng et al. [22] found that *JAZ1–3* in combination with *MYC2–1* suppressed the transcription of *CmFT*, whereas *MYC2–1* alone could enhance the expression of *FT*. Although some scholars have reported the mechanisms involved in the development of catkins in chestnut, this mechanism is still not clear. Therefore, to better understand the flowering mechanism in *Castanea*, we observed the morphological changes and measured the levels of six hormones in male and female flowers at various development stages. We quantified the effects of exogenous ZT and JA applications on flower differentiation. We also analyzed key phytohormone genes using RNA-Seq and RT-qPCR.

## 2. Materials and Methods

### 2.1. Plant Materials and Growth Conditions

Catkins of *C. mollissima* 'Tanqiao' were collected from the second and higher bearing branches from April to late May between 2019 and 2021 at the Changsha chestnut germplasm resources garden (29°2′5″ N, 110°14′18″ E) in Changsha City, Hunan Province, China [23]. The mixed catkins were harvested from eight-year-old *C. mollissima* 'Tanqiao'. We selected six stages with differences in appearance of the mixed catkins collected, with

the first two stages named B0 and B1. For the last four stages, the male and female flowers had differentiated on the mixed catkins, so we separated the male flowers from the upper end of the mixed catkins and named them M1, M2, M3, and M4. The female flowers were separated from the lower end of the mixed catkins and were named F1, F2, F3, and F4 (Figure 1). Parts of the collected catkin samples were used for cytological observations. The other samples were placed in liquid nitrogen and then kept at −80 °C for the determination of endogenous hormone content and RNA extraction.

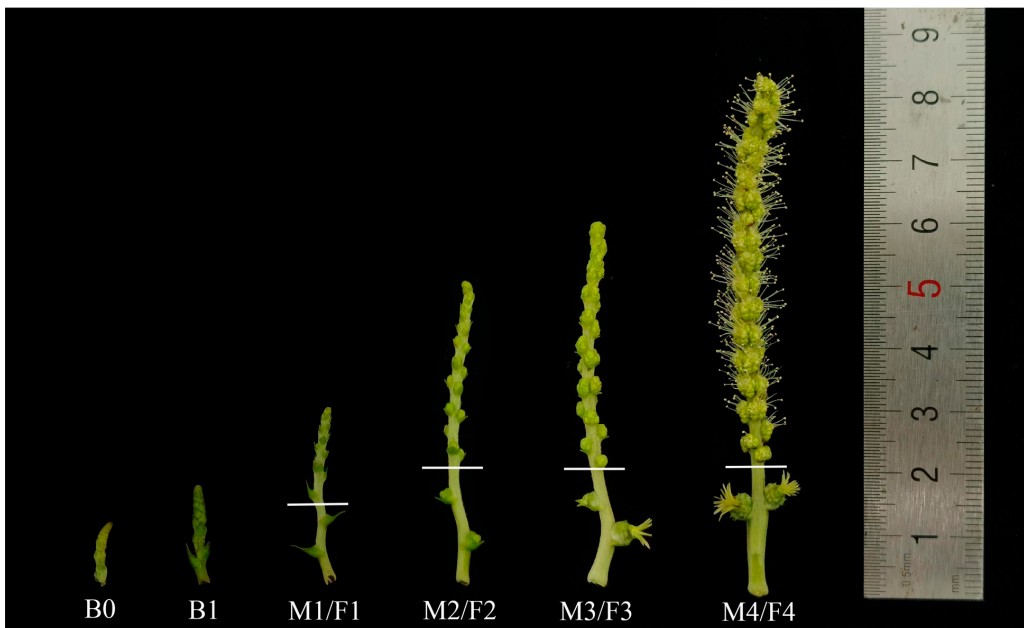

**Figure 1.** Morphological characteristics of the mixed catkins we selected in six different stages. The male flowers differentiate at the upper end of the white straight line, and the female flowers differentiate at the lower end.

### 2.2. Preparation for Paraffin Section

Different phases of male and female flowers were kept in Carnoy's fixative (95% ethanol: acetic acid, 3:1) for approximately 24 h. Samples were dehydrated via successive concentrations of ethanol and xylene and embedded in paraffin (melting point 58~60 °C). Tissues were cut into 6~8 μm slices using a microtome (Leica RM2265., Wetzlar, Germany). Finally, the slices were stained using hematoxylin–eosin or safranine O-fast green. These specimens were observed and photographed under an Olympus microscope [24] (Leica DMi8., Wetzlar, Germany).

### 2.3. Detection of Endogenous Hormones

The mixed catkins from stage B0–B1, the female flowers from F1–F4, and the male flowers from M1–M4 were preserved at −80 °C inliquid nitrogen. These frozen samples were ground to a fine powder. The levels of endogenous hormones ABA, IAA, $GA_3$, jasmonic acid (JA), zeatin (ZT), and brassinosteroid (BR) were extracted with 80% (*v/v*) methanol and detected using high-performance liquid chromatography [25]. Eluted fractions were evaporated, reconstituted with 1 mL 20% (*v/v*) methanol, and injected into a liquid chromatography–electrospray ionization–tandem mass spectrometry apparatus (6410; Agilent, Santa Clara, CA, USA). The standard liquor was diluted to 0.1, 10, 25, 50 and 100 ng/mL standard solution and passed through a 0.45 μm organic filter membrane to draw a standard curve. All measurements were conducted in three biological replicates.

### 2.4. Spray Verification in the Field

The effects of ZT and JA on the flowering of *C. mollissima* were verified using a completely randomized design. Specific amounts of ZT (5 mg·L$^{-1}$, 20 mg·L$^{-1}$) and JA (10 mg·L$^{-1}$, 50 mg·L$^{-1}$) (Solarbio, Beijing, China) were dissolved in 5 mL glacial acetic acid and 5 mL methanol, respectively, for hormone stock. Sixty trees were sprayed three times starting in late March at one-week intervals between sprays. The number of female flowers, male flowers, and mixed catkins; the length of mixed catkins and fruiting branches; and the diameter of fruiting branches were measured at the flowering stage.

### 2.5. RNA Extraction, Sequencing, Quality Control and Functional Annotations

Total RNA was extracted from ten samples with three biological replicates. The mRNA was extracted through oligo (dT) beads, fragmented, and reverse-transcribed to the first-strand cDNA using random hexamers. The Illumina HiSeq 6000 sequencing platform (Illumina, San Diego, CA, USA) by Majorbio (Shanghai, China) was used to sequence the cDNA library. Reads were assembled using Trinity, and assembly integrity was further assessed using BUSCO4.1.4. HISAT2.2.1 was used to align clean reads to the chestnut reference genome [26]. All unigenes were aligned to four protein databases, namely Swiss Prot (UniprotKB) [27], Nr (NCBI Non-Redundant Protein Sequences) [28], KEGG (KEGG Ortho Database) [29], and COG (Clusters of Orthologous Groups) [30].

### 2.6. Analysis of Transcriptomic Data

For each sample, expression levels were estimated as fragments per kilobase of transcripts per million mapped reads (FPKM). DESeq was used to identify DEGs among two samples via FDR < 0.05 and absolute log$_2$ (fold-change) $\geq$ 1 as the limit [31]. The newly acquired DEGs were further annotated with KEGG pathway analysis. Gene co-expression from network analysis was constructed using the WGCNA algorithm [32] by clustering the differential expression of the same or similar genes, and time-series analysis was based on STEM [33]. The significant trend *p*-value was set to 0.05, and the number of time-series patterns was set to 24.

### 2.7. Verification of Transcriptome Data Using RT-qPCR

In order to validate that the RNA-sequencing data were reliable, 6 candidate genes were randomly chosen, and the expression of the DEGs in ten samples was confirmed using RT-qPCR with three biological replicates. Gene expression was calculated using the $2^{-\Delta\Delta Ct}$ method [34], and *ACTIN* was used as the reference gene [22]. The primers for selected genes were designed by Primer (version 5.0) software. Table 1 presents a list of each gene's primer sequence.

**Table 1.** List of primers sequence of selected genes.

| Gene Name | Forward Primer (5′–3′) | Reverse Primer (5′–3′) |
| --- | --- | --- |
| *AGL2* | GTCTGAAGCGCATACGAACA | GTGTGCTTGCTCAGGAATGT |
| *DPOD4* | TGCGGAAATTCGACATGAAC | GATTTCTTCAGGCGGGTTCA |
| *RNS3* | AAGAGGCCGTTAGTTTCACCC | AAGCACATCTGCCCTTTGGA |
| *SOC1* | CCGTCGGCATACAAAAGACAC | CTCCCAGGAGTCTCCGTTTT |
| *TM6* | TAGCCCCTCCATCATAACGACAA | CAGACCGTTCAGATCCTCAC |
| *MADS9* | GGTAAGAGGTTGTGGGATGC | TCGGATACTTGAGAGCCCAT |
| *ACTIN* | ATTCACGAGACCACCTACA | TGCCACAACCTTAATCTTCAT |

### 2.8. Statistical Analysis

Microsoft Excel 2022 was used to calculate the raw data of hormones and RT-qPCR. SPSS 24.0 was performed for one-way ANOVA, including a post hoc test (Duncan test, *p* < 0.05). Origin 2019 was used for plotting, and the heatmap of expression analysis was processed using TB-tools 1.109.

## 3. Results

### 3.1. Morphological Changes during the Differentiation of Female and Male Flowers in Castanea mollissima 'Tanqiao'

The differentiation of mixed catkins started in mid-April and ended in late May, which lasted about 45 days. We divided female flower differentiation into eight stages for *C. mollissima* 'Tanqiao' (Figure 2). The bract primordium formed at the base of the mixed catkin (Figure 2A). As the mixed catkin elongated, the lowest 1–2 bracts differentiated into a female flower cluster primordium during B0 (Figure 2B). Two semilunar protuberances, called sepal primordia, were formed at both ends of the apex of the flower cluster (Figure 2C) and gradually surrounded the top of the flower cluster and formed a hemispherical protuberance on its inner side during B1, called the stamen primordium (Figure 2D); this part continued to differentiate into a new round of conical protuberances, and the pistil primordium formed during F1 (Figure 2E). The pistil primordium rapidly elongated and widened, and the style and stigma appeared during F2 (Figure 2F,G). At flowering stage F3, the angle between style and stigma was 45° (Figure 2H), while three ovary cavities formed at the base in F4 (Figure 2I).

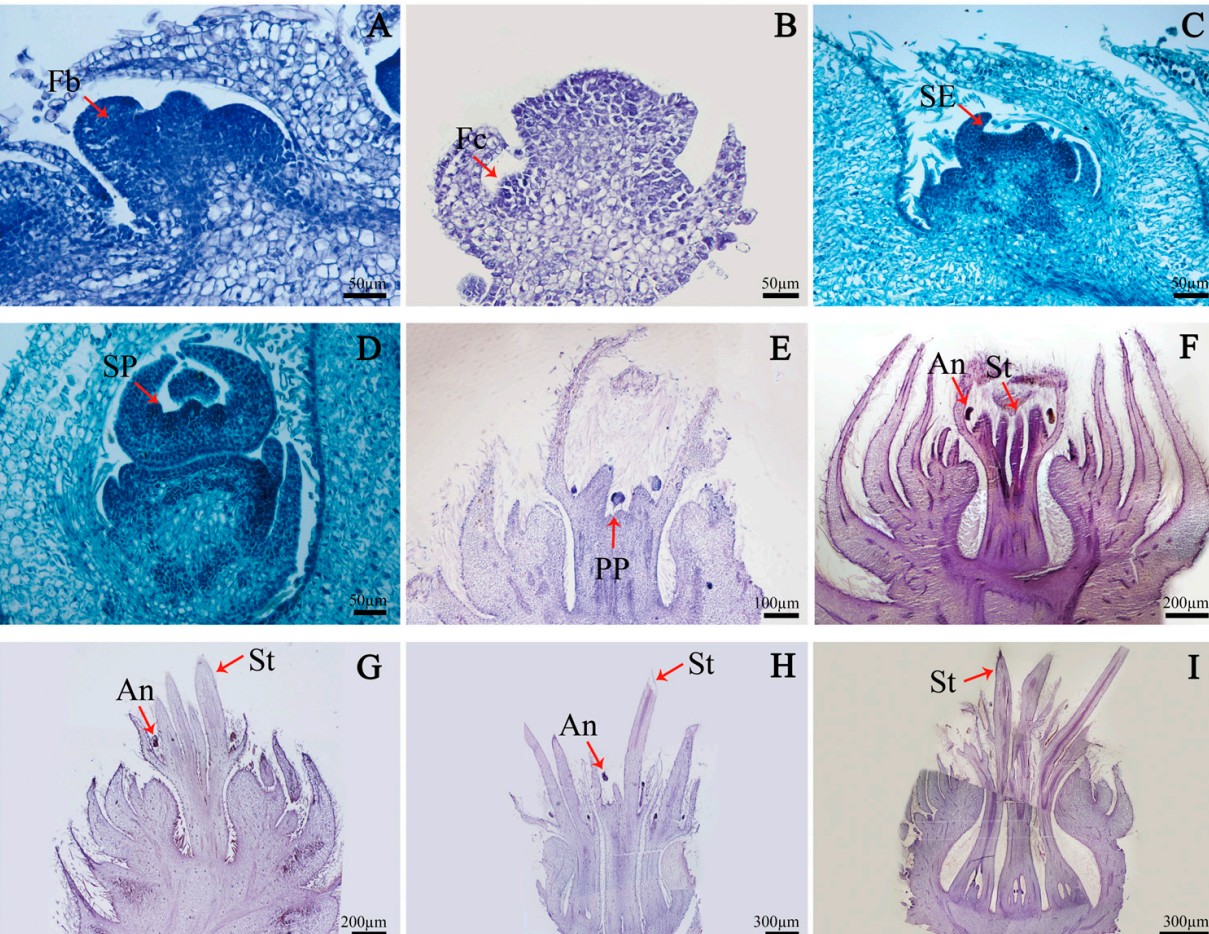

**Figure 2.** Morphological and cytological observation of the differentiation process of female flowers in *Castanea mollissima*. (**A**) Flower-cluster-bract-primordium-differentiation stage; (**B**) flowe-cluster-primordium-differentiation stage (B0); (**C**) sepal-primordium-differentiation stage; (**D**) stamen-primordium-differentiation stage (B1); (**E**) pistil-primordium-differentiation stage (F1); (**F,G**) pistil-development stage (F2); (**H**) flowering stage (F3); (**I**) ovary-formation stage (F4). An: anther; Fb: flower cluster bract primordium; Fc: flower cluster primordium; SE: sepal primordium; SP: stamen primordium; PP: pistil primordium; St: stigma. (**C,D**) are stained with Safranine O-fast green, and others are stained with hematoxylin-eosin.

Compared to the eight stages for a female flower, the male flower was divided into nine stages (Figure 3). Unlike female flowers, after a stamen primordium differentiated into a pistil primordium in M1 (Figure 3E), the pistil primordium developed slowly and gradually degenerated or even disappeared after the formation of the pistil and stamen primordia (Figure 3F). The base of the stamen primordium elongated to form filaments; the top expanded into meristems and then scattered young anthers during M2 (Figure 3G). The anthers gradually matured with two pollen sacs inside during M3 (Figure 3H). Finally, the male flower opened and the pollen was released in M4 (Figure 3I).

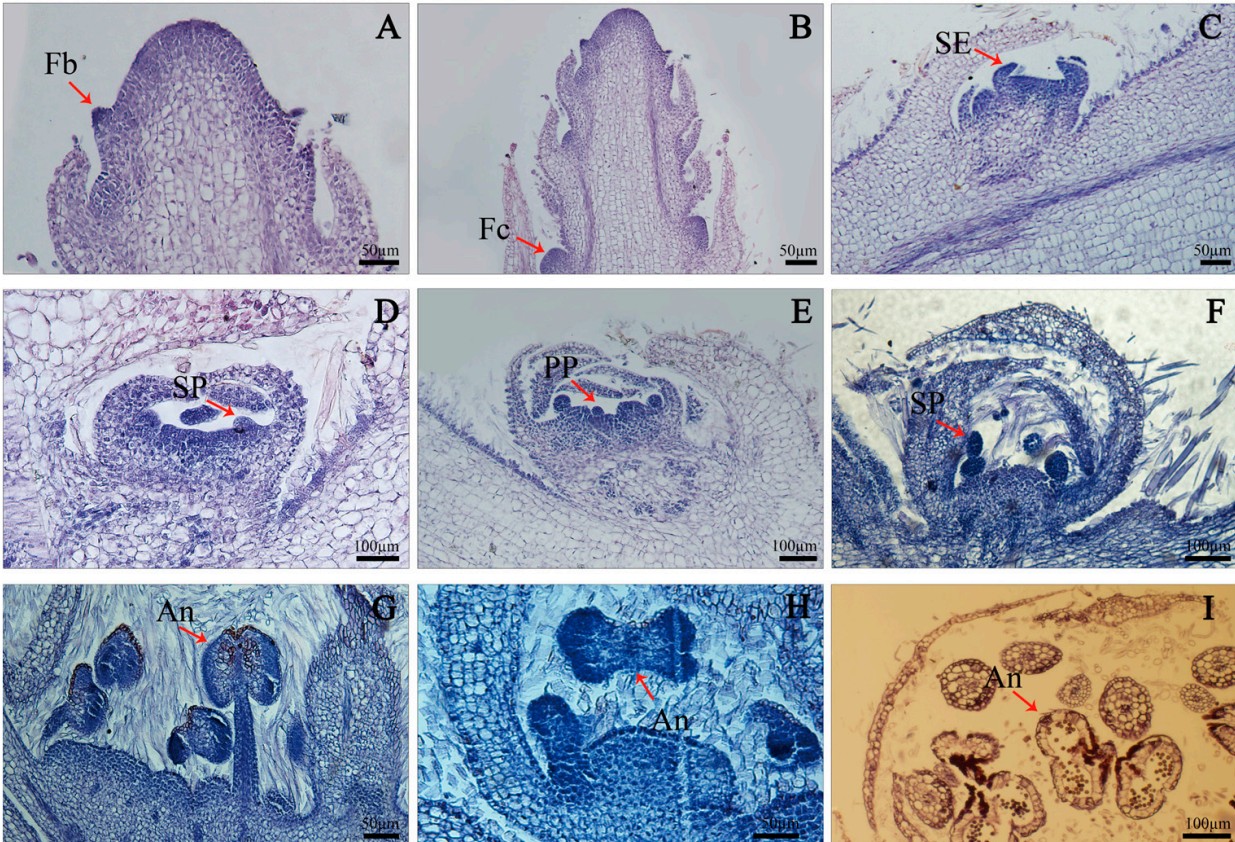

**Figure 3.** Morphological and cytological characterization of the differentiation process of male flowers in *Castanea mollissima*. (**A**) Flower-cluster-bract-primordium-differentiation stage; (**B**) flower-cluster-primordium-differentiation stage (B0); (**C**) sepal-primordium-differentiation stage; (**D**) stamen-primordium-differentiation stage; (**E**) pistil-primordium-differentiation stage (M1); (**F**) stamen-elongation stage; (**G**) anther-formation stage (M2); (**H**) anther-development stage (M3); (**I**) flowering stage (M4). Figure 3I is stained with Safranine O-fast green, and the others are stained with hematoxylin-eosin.

### 3.2. Dynamics of Endogenous Hormones during Female and Male Flower Differentiation in Castanea mollissima 'Tanqiao'

Of the six endogenous hormones examined, $GA_3$ levels were highest at four stages in male flowers, exhibiting a trend of a sharp increase starting at M1 and reaching the highest level at M4 (Figure 4A). As the male flowers developed, ABA was notably higher than those in female flowers and reached a peak of 1216.56 ng/g in the M3 stage (Figure 4B).

In contrast, the levels of JA and ZT in female flowers were higher than those in male flowers during F1 to F4. Both ZT and JA showed a decreasing trend from B0 to B1 and in F1, but an increasing trend in F3 (Figure 4D,E). The level of IAA showed an M-type trend in both male and female flowers (Figure 4C); however, the level of BR showed no significant differences (Figure 4F).

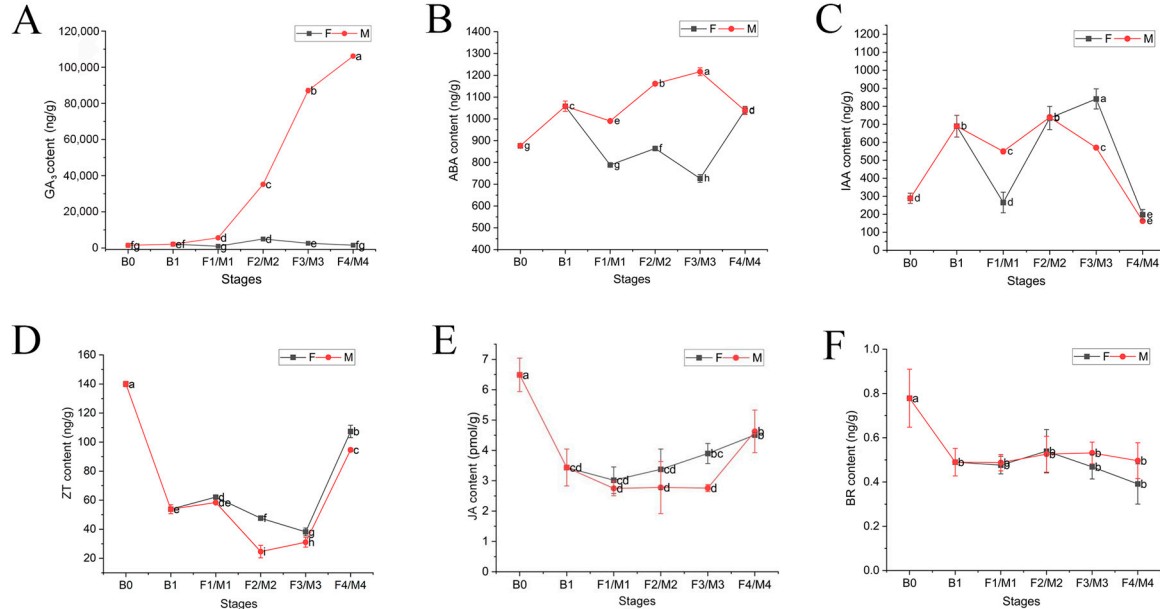

**Figure 4.** Dynamic changes in eight endogenous hormones during the *Castanea mollissima* 'Tanqiao' flower-differentiation process. (**A**) The levels of gibberellic acid (GA$_3$) during the flower-differentiation process of *C. mollissima*; (**B**) the levels of abscisic acid (ABA) during the flower-differentiation process of *C. mollissima*; (**C**) the levels of auxin (IAA) during the flower-differentiation process of *C. mollissima*; (**D**) the levels of zeatin (ZT) during the flower-differentiation process of *C. mollissima*; (**E**) the levels of Jasmonate (JA) during the flower-differentiation process of *C. mollissima*; (**F**) the levels of brassinosteroid (BR) during the flower-differentiation process of *C. mollissima*; F: Female flower; M: Male flower. Standard deviation is symbolized by the error bars. Different letters represent significant difference at *p* < 0.05, *n* = 3.

### 3.3. Spray Verification in the Field

Compared with the control, ZT treatments greatly increased the quantity of mixed catkins and female flowers, but the effect on the fruiting branches was not significant (Figure 5). The best treatment in ZT was 20 mg/L, which significantly increased the quantity of female flowers by 2.53 per single fruit branch compared with the control. However, JA had no significant effect on either sex differentiation or the growth of fruit-bearing shoots (Table 2). This result suggests that ZT exhibited an important regulatory function during the differentiation of female flowers in *C. mollissima*.

**Table 2.** Effects of JA and ZT on flower sex differentiation and the growth of fruit-bearing shoots of *Castanea mollissima*.

| Hormone Type | | Fruit Branch | | | | | |
|---|---|---|---|---|---|---|---|
| Type | Concentration (mg/L) | The Number of Female Flowers on Each Fruit Branch/per | Male Catkin/Strip | Mixed Catkin/Strip | The Length of Mixed Catkin on Each Fruit Branch/cm | The Length of Fruit Branch/cm | The Diameter of Fruit Branch/cm |
| JA | CK | 8.65 ± 2.12 [a] | 8.90 ± 3.18 [a] | 4.46 ± 0.99 [a] | 5.14 ± 1.38 [a] | 21.77 ± 5.82 [a] | 4.15 ± 0.66 [a] |
|  | 10 | 7.88 ± 2.40 [a] | 8.53 ± 2.58 [a] | 4.00 ± 1.18 [ab] | 4.45 ± 1.33 [b] | 20.79 ± 4.18 [a] | 4.14 ± 0.61 [a] |
|  | 50 | 7.63 ± 2.00 [a] | 9.43 ± 3.27 [a] | 3.78 ± 0.97 [b] | 5.01 ± 1.48 [a] | 22.10 ± 4.55 [a] | 4.22 ± 0.63 [a] |
| ZT | CK | 7.71 ± 1.70 [c] | 6.85 ± 2.48 [a] | 3.94 ± 0.89 [b] | 5.87 ± 1.69 [a] | 25.11 ± 6.77 [a] | 4.03 ± 0.68 [a] |
|  | 5 | 9.10 ± 1.97 [b] | 6.20 ± 3.21 [a] | 4.48 ± 1.06 [a] | 5.23 ± 2.10 [ab] | 25.76 ± 8.02 [a] | 3.78 ± 0.62 [a] |
|  | 20 | 10.24 ± 2.71 [a] | 7.20 ± 2.48 [a] | 4.89 ± 1.22 [a] | 5.05 ± 1.78 [b] | 24.84 ± 5.75 [a] | 3.82 ± 0.48 [a] |

Data are expressed as mean ± standard deviation, ± means plus or minus; different letters in the same column indicate significant differences at a 0.05 level, while the same letter indicates no significant difference.

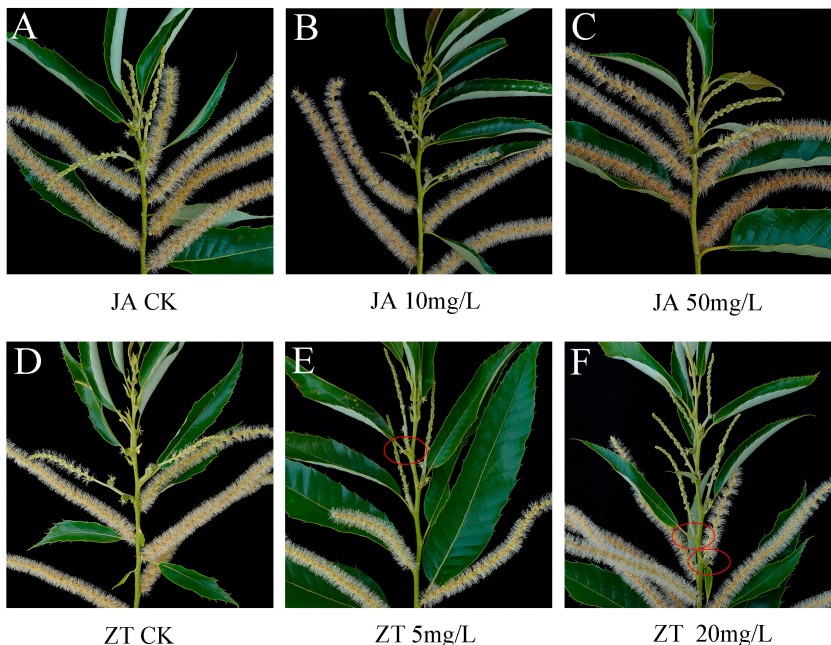

**Figure 5.** The mixed catkin of *Castanea mollissima* under different treatments. (**A**,**D**) represent the control treatment of jasmonic acid and zeatin, respectively; (**B**,**C**) represent the application of 10 mg and 50 mg jasmonic acid, respectively; (**E**,**F**) represent the application of 5 mg and 20 mg zeatin, respectively; CK means the control group of treatment with pure water. The axillary female flowers are circled with red circles.

### 3.4. Basic Data of Transcriptome Sequencing

After filtering the raw data, each sample's high-quality clean readings reached more than 98%. The percentage of Q20 base and Q30 base was greater than 96% and 92%, respectively. The percentage of GC ranged from 42.21% to 45.37%. These reference data suggest that the sequencing results were trustworthy (Table S1). The reference sequence for comparison was the chestnut genome [35], and the results of the comparison are presented in Table S2.

We annotated All-Unigene in the six databases: the Gene Ontology (GO) database, the KEGG protein database (KEGG), the Clusters of orthologous groups (COG) database, NCBI's Non-redundant protein database (NR), the Swiss Prot protein database (Swiss Prot), and the Pfam protein families database (Pfam). A total of 34,053 Unigene sequences were annotated. The NR database had the most annotations, accounting for 94.96% of the whole, while the KEGG database contained the fewest annotations, accounting for 37.50% (Table 3).

**Table 3.** The statistical results of functional annotations in *Castanea mollissima* 'Tanqiao'.

| Database | Number of Annotated Genes | Percentage of Annotated Genes (%) | Percentage of All-Unigene (%) |
| --- | --- | --- | --- |
| GO | 22,244 | 63 | 61.91 |
| KEGG | 13,474 | 38.03 | 37.50 |
| COG | 27,346 | 77.55 | 76.11 |
| NR | 34,117 | 94.77 | 94.96 |
| Swiss Prot | 24,861 | 70.52 | 69.20 |
| Pfam | 25,265 | 71.84 | 70.32 |
| Total | 34,053 | | |

### 3.5. Differentially Expressed Genes during Female and Male Flower Differentiation

A total of 20,510 differentially expressed genes were found during mixed catkin differentiation, of which 10,302 were upregulated and 10,208 were downregulated. In M4 vs. F4, the most DEGs were discovered, with 2264 upregulated genes and 2950 downregulated genes, respectively, while F1 vs. F2 generated the fewest DEGs (Figure 6A,B). The number of timepoint-specific DEGs in male flowers ranged from 310 (M1 vs. M2) to 3038 (M3 vs. M4), while in female flowers, the number ranged from 77 (F1 vs. F2) to 1868 (B0 vs. B1). Furthermore, throughout the five comparisons, 55 and 6 genes exhibited substantially distinct expression in male and female flowers, respectively (Figure 6C,D).

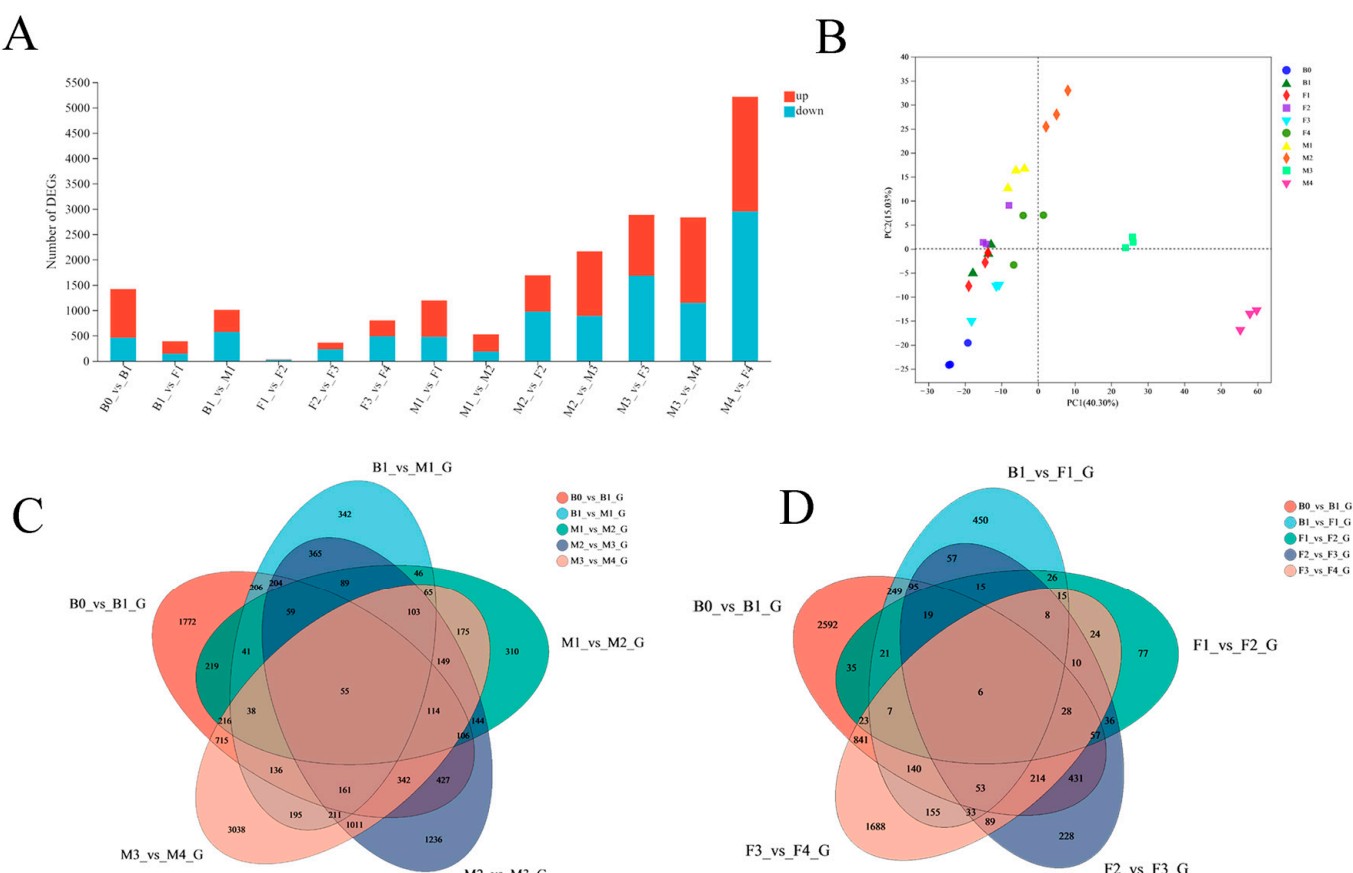

**Figure 6.** (**A**) The number of differentially expressed genes (DEGs) in different comparisons. Up− and down−regulated DEGs are shown in the orange and blue columns, respectively. (**B**) The PCA analysis of all groups; (**C**) Venn diagrams of DEGs in male flowers; (**D**) Venn diagrams of DEGs in female flowers.

### 3.6. Analysis of KEGG Pathway of DEGs

The DEGs in male flowers were mapped to 1, 9, and 13 KEGG pathways that were significantly enriched in M1 vs. M2, M2 vs. M3, and M3 vs. M4, respectively. The most common enriched pathways in male flowers were phenylpropanoid biosynthesis, plant hormone signal transduction, and flavonoid biosynthesis (Figure 7A). In comparison with male flowers, the DEGs in female flowers were mapped to 13, 12, and 10 KEGG pathways that were significantly enriched in F1 vs. F2, F2 vs. F3, and F3 vs. F4, respectively. Within these important pathways, starch and sucrose metabolism occurred in all comparisons, while zeatin biosynthesis, ABC transporters, phenylpropanoid biosynthesis, and protein processing in the endoplasmic reticulum occurred in two comparisons. Furthermore, the MAPK signaling pathway-plant was highly enriched in F3 vs. F4 (Figure 7B).

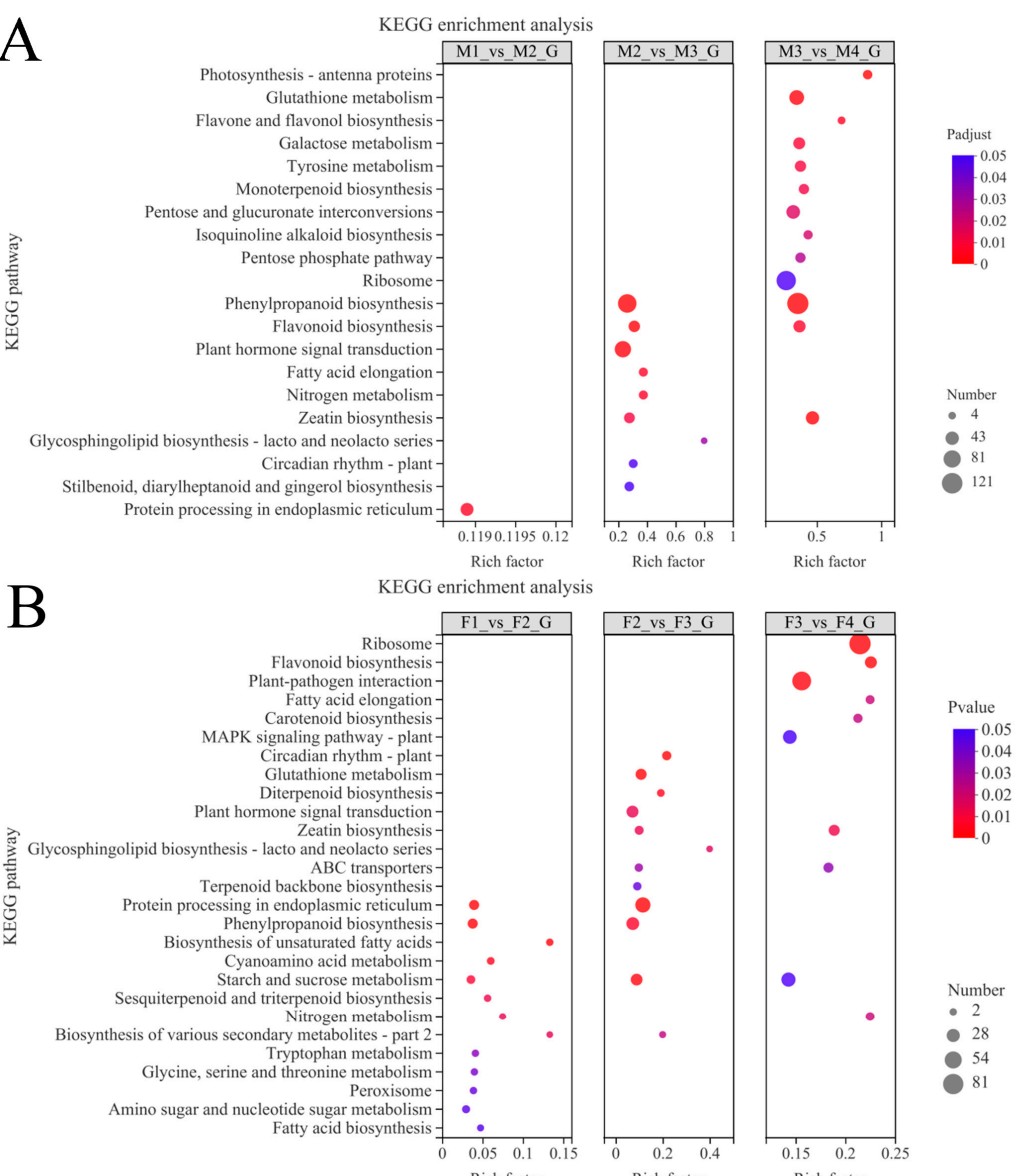

**Figure 7.** (**A**) The significant KEGG enrichment pathway of DEGs in male flowers at different developmental stages. (**B**) The significant KEGG enrichment pathway of DEGs in female flowers at different developmental stages. The area of the bubbles represents the quantity of enriched DEGs, while the color of the bubbles symbolizes the *p*-adjust and *p*-value shown in the panel to the right.

### *3.7. WGCNA Analysis*

To reveal the key genes associated with the development of female flowers, we ran the RWGCNA package on all 30 samples. A soft-threshold power of eighteen was inserted into the network topology to uncover the network's scale independence and mean connectedness (Figure 8A,B).

The dynamic shear method was used to divide the modules, and a total of 13,574 genes were clustered into 22 modules with a similarity greater than 75%. The turquoise module contained the largest number of genes at 3850, while the dark module had the smallest number at 45. Using an absolute value of Pearson correlation coefficient >0.7 and *p* < 0.05 as the screening condition, three modules specifically related to female flower development were identified. The pink module (R = 0.775) was positively correlated with female flower development (Figure 8C,D). Thus, we mapped the gene co-expression network of the pink module; among these 10 genes (Table 4), the transcription factor *IDD7* may be the key gene during the female flower differentiation of *C. mollissima* 'Tanqiao' (Figure 8E).



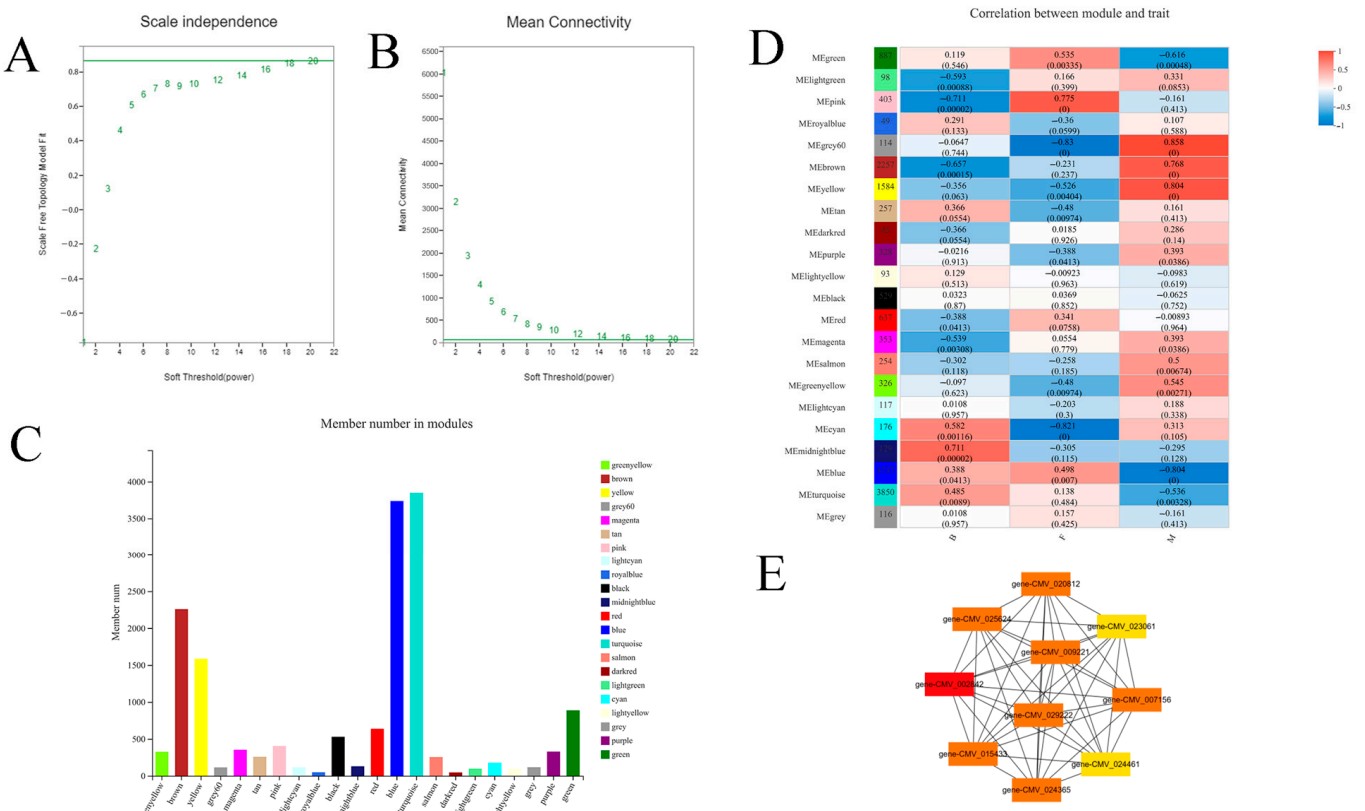

**Figure 8.** (**A**) Fitting index R$_2$ of the scale-free network corresponding to different soft thresholds; (**B**) the average connectivity corresponding to different soft thresholds; (**C**) the module gene number distribution; and (**D**) correlation heat map between modules and traits. The data outside the brackets and the data inside the brackets represent the correlation coefficient and p-value among modules and traits, respectively. (**E**) Core gene co-expression network.

**Table 4.** Functional annotations of hub genes in the pink module.

| Gene ID | Gene Name | Functional Annotation |
| --- | --- | --- |
| gene-CMV_002842 | *IDD7* | Protein indeterminate-domain 7 |
| gene-CMV_007156 | *BBX19* | B-box zinc finger protein 19 |
| gene-CMV_009221 | *APC1* | Anaphase-promoting complex subunit 1 |
| gene-CMV_015433 | *EF1G3* | Elongation factor 1-gamma 3 |
| gene-CMV_020812 | *ARATH* | Pentatricopeptide repeat-containing protein |
| gene-CMV_023061 | *DIRL1* | Putative lipid-transfer protein DIR1 |
| gene-CMV_024365 | *ACD6* | Protein ACCELERATED CELL DEATH 6 |
| gene-CMV_024461 | *NAKR1* | Protein SODIUM POTASSIUM ROOT DEFECTIVE 1 |
| gene-CMV_025624 | *UFOG5* | Anthocyanidin 3-O-glucosyltransferase 5 |
| gene-CMV_029222 | *LBD25* | LOB domain-containing protein 25 |

### 3.8. DEGs Related to Flower Differentiation

According to the results of WGCNA, transcription factors exhibit an essential role in the differentiation of the female and male flowers of *C. mollissima*. Based on 266 common DEGs between four comparative groups (Figure 9A), a total of 16 transcription factors related to the differentiation of female and male flowers were excavated. Among the 16 transcription factors, the expression levels of *bHLH52*, *AHL23*, *ERF084*, and *VRN1* were downregulated with flower development, while *bHLH92*, *WRKY40*, and *HSP17.7* were upregulated in the early stages of female flower development. The expression levels of *KUA1*, *MYB26*, and *LAX1* were significantly high during F1 to F4 and decreased from M1, while *NAC056*, *AG*, *MADS9*, and *AMS* were highly increased during the male flower stages (Figure 9B).

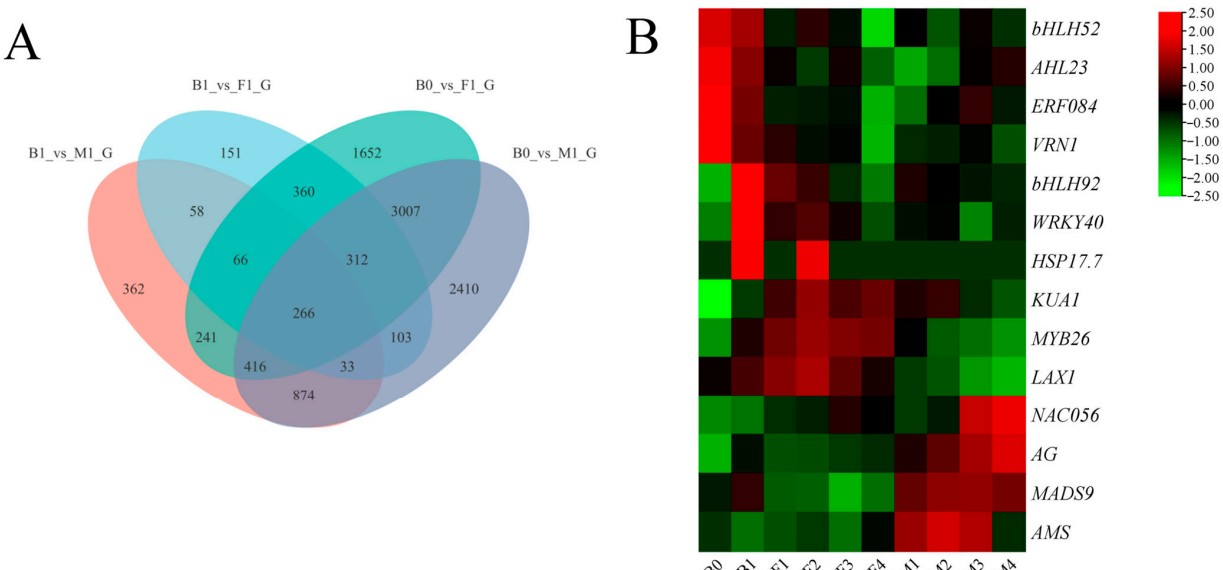

**Figure 9.** (**A**) Venn diagrams of DEGs between mixed inflorescence and female and male flowers at differentiation stages in *Castanea mollissima* 'Tanqiao'. (**B**) Heatmap of DEGs related to the transcription factor in *C. mollissima* 'Tanqiao'. Red and green represent upregulated and downregulated, respectively.

*3.9. Time-Series Analysis*

The transcriptome data of six stages of B0 to F4 and B0 to M4 were analyzed using the STEM time-series clustering algorithm. The expression trends of profile0 and profile23 decreased and increased during the development of male and female flowers, respectively (Figure 10A,B). Combined with the dynamic changes in endogenous hormone contents in the development of male and female flowers, three genes related to JA biosynthesis and signal transduction and 34 genes related to ZT biosynthesis and signal transduction were screened in profile0 and profile23 (Table 5).

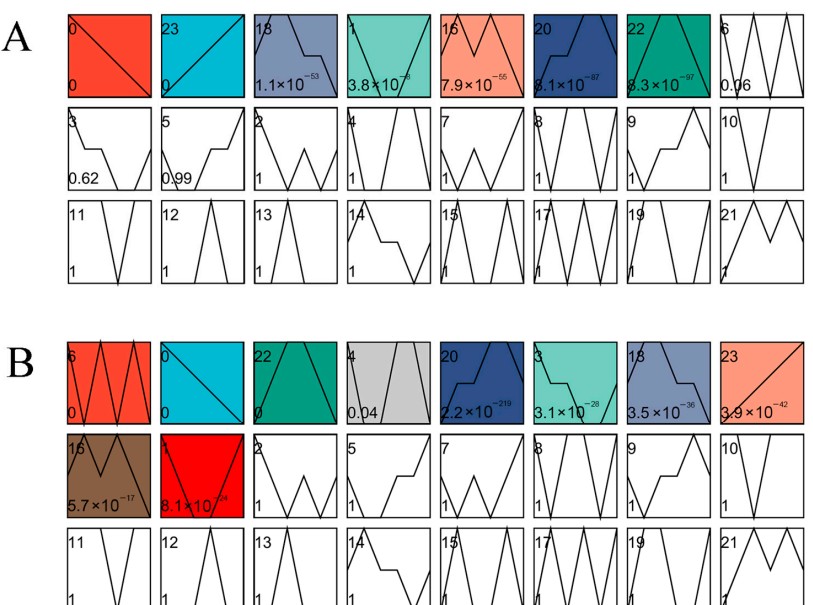

**Figure 10.** (**A**) Gene expression profiling of all genes through 6 stages of male flowers. (**B**) Gene expression profiling of all genes through 6 stages of female flowers. The temporal pattern of profiles with colors is conspicuous, while the temporal pattern of profiles without colors is not conspicuous. The number of the profile is shown in the upper-left corner of the rectangle, the number in the lower-left corner is the *p*-value, and the trend of expression over time is symbolized by the line.

**Table 5.** Functional annotations of significant genes related to ZT and JA in the 0.3 profile in *Castanea mollissima* 'Tanqiao'.

| Gene Pathway Definition | Gene Name | Number | Gene ID |
|---|---|---|---|
| JA biosynthesis and signal transduction | *AOS3* | 1 | gene-CMV_027313 |
| | *COIL* | 1 | gene-CMV_028222 |
| | *MYC2* | 1 | gene-CMV_022981 |
| Zeatin biosynthesis and signal transduction | *AHK3* | 1 | gene-CMV_002656 |
| | *ARR17* | 1 | gene-CMV_030552 |
| | *UGTK4* | 1 | gene-CMV_017378 |
| | *UFOG4* | 1 | gene-CMV_008253 |
| | *CKX5* | 1 | gene-CMV_017974 |
| | *CKX3* | 1 | gene-CMV_014383 |
| | *CKX6* | 1 | gene-CMV_022840 |
| | *LOGL1* | 1 | gene-CMV_018548 |
| | *LOG1* | 2 | gene-CMV_025587 |
| | | | gene-CMV_005650 |
| | *LOG3* | 1 | gene-CMV_020510 |
| | *LOG5* | 2 | gene-CMV_017862 |
| | | | gene-CMV_009202 |
| | *UGT8* | 2 | gene-CMV_014086 |
| | | | gene-CMV_026607 |
| | *7DLGT* | 2 | gene-CMV_010956 |
| | | | gene-CMV_026588 |
| | *U73C6* | 4 | gene-CMV_008252 |
| | | | gene-CMV_014625 |
| | | | gene-CMV_015292 |
| | | | gene-CMV_015294 |
| | *UGTK5* | 5 | gene-CMV_017379 |
| | | | gene-CMV_021009 |
| | | | gene-CMV_017377 |
| | | | gene-CMV_017576 |
| | | | gene-CMV_027806 |
| | *UGT2* | 8 | gene-CMV_005081 |
| | | | gene-CMV_008238 |
| | | | gene-CMV_023426 |
| | | | gene-CMV_025853 |
| | | | gene-CMV_027908 |
| | | | gene-CMV_023427 |
| | | | gene-CMV_006423 |
| | | | gene-CMV_021488 |

### 3.10. DEGs Associated with Zeatin and Jasmonic Acid during Flower Differentiation

According to the results of KEGG enrichment analysis and time-series analysis, we extracted 3 genes related to jasmonic acid biosynthesis and signal transduction, 3 genes related to zeatin biosynthesis, and 18 genes related to zeatin signal transduction. The expression levels of *LOG5*, *ARR4*, *ARR9*, and *UFOG4* were highly expressed from F1 to F4, while *CKX5/6* and *7DLGT* were highly expressed from M3 to M4. Interestingly, the gene family *LOG*, which is related to zeatin biosynthesis, was only highly expressed in female flowers, suggesting that gene family *LOG* may be the core hormone gene during female flower differentiation (Figure 11A).

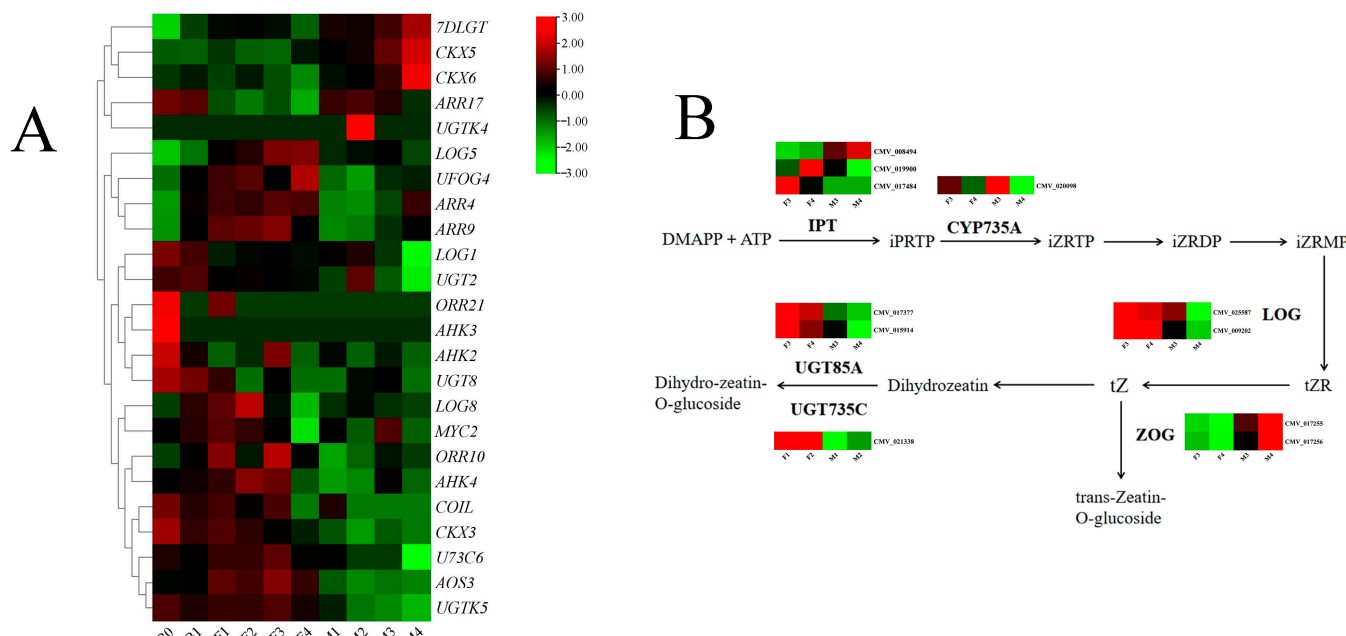

**Figure 11.** (**A**) Heatmap of DEGs related to jasmonic acid and zeatin in *Castanea mollissima* 'Tanqiao'. Red and green represent up- and downregulated. (**B**) Heatmap of DEGs related to the zeatin biosynthesis pathway. iPRTP: Isopentenyl-ATP; iZRTP: trans-Zeatin riboside triphosphate; iZRDP: trans-Zeatin riboside triphosphate; iZRMP: trans-Zeatin riboside monophosphate; tZR: trans-Zeatin riboside; tZ: trans-Zeatin.

In the ZT biosynthesis pathway, *IPT*, *CYP735A*, and *LOG* were the main enzymes that synthesize ZT. As an early gene in ZT biosynthesis, *IPT* was upregulated during M3-M4 and expressed in F3 and F4 in female flowers. Cytokinin hydroxylase *CYP735A* was upregulated at M3 and then sharply downregulated at M4. *LOG*, coding for a cytokinin-activating enzyme in the later stage of ZT biosynthesis, was only highly expressed in F3 and F4 in female flowers. These findings indicate that cytokinin activity may be higher in female flowers. Downstream of zeatin biosynthesis, cytokinin glycosyltransferase *UGT85A* and *UGT735C* were only expressed in female flowers, while *ZOG* was only expressed in male flowers. These findings suggest that, as flowers differentiated, the expression of the genes involved in ZT biosynthesis and pathways changed (Figure 11B).

*3.11. RT-qPCR Verification*

RT-qPCR experiments were performed on the ten samples (B0, B1, F1, F2, F3, F4, M1, M2, M3, and M4) to validate the reliability of the transcriptome data. Thus, we randomly selected six genes in transcriptome data. As shown in Figure 12, the expression patterns of the six genes obtained from RT-qPCR were almost consistent with those from RNA-seq, demonstrating that the RNA-seq results were credible.

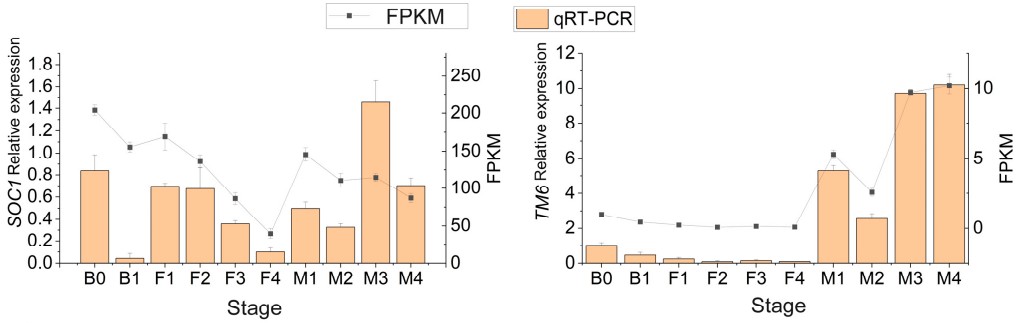

**Figure 12.** *Cont.*

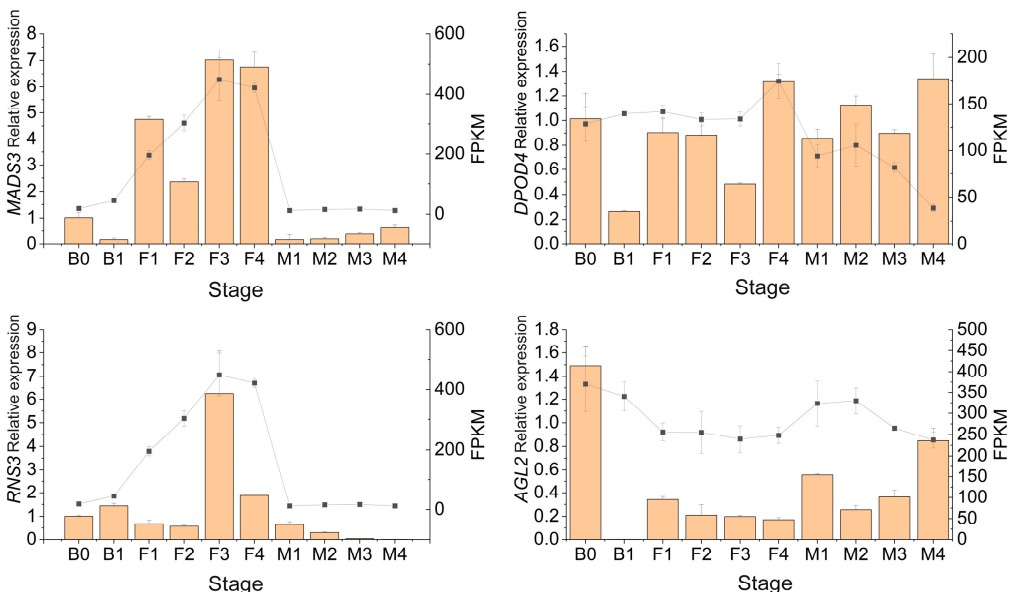

**Figure 12.** RT-qPCR validation of the expression patterns of 6 differentially expressed genes (DEGs) during male and female flower development. The x-axis represents the 10 stages; the left y-axis represents the relative expression levels normalized from the RT-qPCR experiments; and the right y-axis represents the FPKM values from RNA-seq data. Error bars represent the standard deviation (SD) of the mean (*n* = 3).

## 4. Discussion

### 4.1. The Differentiation Process of Male and Female Flowers in Castanea mollissima

As a monoecious plant, *C. mollissima* has two types of catkins: pure male catkins and mixed catkins. The upper part of the mixed catkins differentiates into male flowers, while the lower part differentiatess into female flowers. Since the female flower is the foundation of yield in *C. mollissima*, it is essential to study the process of mixed catkin differentiation in production practice. Some scholars observed the process of male flower-bud differentiation in Chinese chestnut. Chen et al. [36] found that female flower formation includes the development of the flower primordium, sepal primordium, pistil primordium, style, stigma, ovary, ovule, and embryo sac. Yan et al. [37] showed that the primordia of bisexual catkins appeared in early March, while the primordia of female flower clusters became ovule primordia in mid-May for 'Yongfeng No.1'. In our paper, we subdivided the primordia into different types: flower cluster bract primordium, flower cluster primordium, sepal primordium, stamen primordium, and pistil primordium. In female flowers, the inner side of the stamen primordium differentiated into pistil primordia and then rapidly elongated and widened and became stigma. Unlike female flowers, pistils developed slowly and gradually degenerated or even disappeared after the formation of the pistil primordia in male flowers. Thus, we speculate that pistil primordium differentiation stage is critical in mixed catkin differentiation.

### 4.2. Dynamic Change in Endogenous Hormones during Female and Male Flower Differentiation

Numerous studies have proven that plant hormones are involved in flower differentiation. Plant hormones, including GA, ABA, and IAA, among others, had a significant impact on flower differentiation. The impact on flowering varied between plants [38]. High levels of IAA and ABA were beneficial for female flower differentiation in *Zanthoxylum planispinum* var. Dingtanensis [39]. Low levels of IAA can promote flower differentiation in *Camellia perpetua* and *Ziziphus jujuba* Mill. during early stages, while high levels of IAA are required during late stages [40,41]. In our study, the IAA level in female flowers was higher than in male flowers from the F2 to F4 stages and showed a similar trend. The accumulation of ethephon (CEPA) and ABA could inhibit female differentiation in *C. henryi* [42]

and other species, such as *Eriobotrya japonica Lindl* 'Ninghaibai' [43] and *Manihot esculenta* Crantz [44]. Similar results were observed in *C. mollissima*, where the high contents of $GA_3$ and ABA promoted male flower differentiation, but the dynamic changes in ZT and JA were favorable for female flower differentiation. ZT is one type of CK, and similar research has reported that CK plays an important role in the flower development of *C. henryi* [45].

### 4.3. Exogenous ZT Application Alters the Number of Female Flowers

In order to enhance the yield of monoecious plants, increasing the number or ratio of female flowers is necessary. Exogenous IAA [46], 6-BA [47], and $GA_3$ [10] were demonstrated to enhance the number of female flowers in various plants. CK has been discovered to increase the quantity of female flowers significantly and even had a feminization effect on male flowers. A 50 mg/L forchlorfenuron (CPPU) treatment could turn the male plant into a bisexual plant in *Vitis amurensis* [48]. A 125 mg/L CPPU can transform a male catkin into a mixed catkin in *C. henryi*, but the transformed catkins were sterile [49]. ZT is a natural plant CK found in vascular plants. Our results showed that 20 mg/L ZT significantly increased the quantity of fertile female flowers compared with the control. This suggests that an amazing improvement in the quantity of female flowers might be achieved with the appropriate application of ZT.

### 4.4. Transcription Factors and Genes Related to Zeatin Co-Regulate the Flower Differentiation

Numerous studies have reported that phytohormones participate in the flower differentiation of many plants. Among numerous phytohormones, CK plays a key role in influencing flower differentiation between males and females. In sweet cherry, the regulatory network of female flowers is related to zeatin [50]. Some B-type *MdRR* genes (*MdRRB9* and *MdRRB11*) are implicated in the positive impact of CK on apple flower induction [51]. Furthermore, CK could change the fate of the apical meristem in male flowers and stimulate the growth of carpel primordia in *P. volubilis* [52]. CPPU accelerated the development of pistil and induced the maturation of the octonucleate embryo sac in male flowers, which induced a sex change in *Vitis amurensis* Rupr. with fertility [53], which is consistent with CK treatment in *Jatropha curcas* [54]. ZT, as one type of CK, isopentenyl transferase (*IPT*); cytochrome P450 monooxygenase, family 735, subfamily A (*CYP735A*); and cytokinin-activating enzyme (*LOG*) were the main enzymes that synthesize ZT. In *Arabidopsis*, transgenic plants with the AP1::IPT4 gene significantly increased the number of flower primordia, mediating the upregulation of *CUC3* and *LBD3* in the development of inflorescence [55]. *LOG* encodes a cytokinin enzyme that works in the final step of cytokinin synthesis. It can convert inactivated cytokinin nucleotides into free-base forms with biological functions directly through its specific phosphoribohydrolase. *LOG1/3/7*, which is related to ZT biosynthesis, joins in the determination of flower sex in *C. henryi* [49]. The conclusions of the present research on *C. mollissima* are a further proof of these findings, as *LOG1/5/8* were highly expressed in female flower differentiation. Thus, we speculate that the number of female flowers highly correlates with the activity of ZT in *C. mollissima*, and the *LOG* gene family might be the core hormone gene during female flower differentiation.

Transcription factors (TFs) perform an essential role in plant growth and development, particularly in flower differentiation [56]. Members of the *MYB* transcription factor have been extensively studied in many plant species. DEGs related to MYB families involved in the process of flower differentiation were found in *Schisandra chinensis* [57], *Prunus avium* L. cv. *Bing* [50], and *Idesia polycarpa* Maxim. var vestita Diels [58]. The *MR1TCONS_00020658.1* was only expressed in male flowers of red bayberry [59]. In the late stages of female flower development in *Idesia polycarpa* Maxim. var vestita Diels, the downregulation of the B gene family had an impact on stamen fertility [58], which suggests a vital role in flower differentiation. Meanwhile, the *bHLH* family, such as *PAVMYC2*, was also proven to be essential in the flower differentiation of *Osmanthus* [60]. The transcription level of *PpIDD11* in peach was highest in the pistil, and its overexpression mutant exhibited an abnormal stretch of the stigma [61]. Previous research has shown that transcription factors could

regulate flower differentiation with phytohormone [47,62]. PpIDD-DELLA1 complexes activated the transcription of *PpGA20ox1* [63], while exogenous GA induced the downregulation of *MDIDD7* to regulate flowering in apple 'Changfu 2' [64]. According to our results, we found that some TFs, including *bHLH92*, *IDD7*, and *MYB26* genes, are involved in the flower development of *C. mollissima*. These findings suggest that transcription factors and their interactions with phytohormones might regulate flower differentiation.

## 5. Conclusions

This study explored the mechanisms involved in the differentiation of mixed catkins in *C. mollissima* 'Tanqiao'. The hormone levels of JA and ZT were highly related to female flower development. Through KEGG pathway analysis, time-series analysis, and WGCNA analysis, 21 genes related to zeatin biosynthesis and transduction and 16 transcription factors were screened. Among these genes, we speculate that the *LOG* gene family and *IDD7* may be the core regulatory factors in regulating female flower development. But whether *LOG* interacts with *IDD7* remains to be further studied. In addition, spraying zeatin could significantly increase the number of fertile female flowers. These findings are critical for understanding the mechanism of mixed catkin development in Chinese chestnut.

**Supplementary Materials:** The following supporting information can be downloaded at https://www.mdpi.com/article/10.3390/f14102057/s1. Supplementary Table S1: Statistics of raw reads for transcriptome in *C. mollissima* 'Tanqiao'. Table S2: Data of clean reads mapping to the reference chestnut genome in each stage.

**Author Contributions:** Conceptualization, X.Z. and L.W.; methodology, X.Z., L.W., Q.Y. and X.L.; formal analysis, X.Z. and L.W.; validation and investigation, D.Y. and H.X.; writing—original draft, X.Z.; writing—review and editing, X.Z., F.Z. and J.M. All authors have read and agreed to the published version of the manuscript.

**Funding:** This study was financially supported by the National Key R&D Program of China (Grant No. 2022YFD2200400), Scientific research project of Hunan Provincial Department of Education (Grant No.21A0180), and Central Finance Forestry Science and Technology Promotion Demonstration Fund Project (Grant No. [2021] XT01).

**Data Availability Statement:** The data presented in this study are available on request from the corresponding author. The data are not publicly available due to privacy.

**Conflicts of Interest:** The authors declare no conflict of interest.

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
