# Peer review of "Transcription Factor and Zeatin Co-Regulate Mixed Catkin Differentiation of Chinese Chestnut (Castanea mollissima)"

_forests, doi:10.3390/f14102057_

Round 1

Reviewer 1 Report

The manuscript primarily deals with flower development and differentiation in the monoecious fruit crop Castanea mollissima. The genetics of flower development in plants have been the subject of much research in the last decade; however, understanding the mechanism of mixed catkin development is an important but difficult subject. Although there are few earlier reports on flower development in Castanea, the authors have dealt with both male and female flower development in detail, intending to improve the imbalance of male-to-female flowers for chestnut production.

The morphological details of flower development ( male and female ) in both Material and Method and results can be improved. Figures of Catkins, male and female flowers, accompanied by their histology, would enable readers to understand the stages of development better. This is an interesting work.

Author Response

Dear Reviewer,

Thank you very much for your time involved in reviewing the manuscript and your very encouraging comments on the merits. We also appreciate your clear and detailed feedback and hope that the explanation has fully addressed all of your concerns. In the remainder of this letter, we discuss each of your comments individually along with our corresponding responses. To facilitate this discussion, we first repeat your comments and then present our responses to the comments.

Comments1: The morphological details of flower development (male and female) in both Material and Method and results can be improved. Figures of Catkins, male and female flowers, accompanied by their histology, would enable readers to understand the stages of development better.

Response1: We agreed with this comment. The morphology of flower development (male and female) had been detailed in both material and method and results. Figures of Catkins, male and female flowers were also added. Please see the Figure 1. Morphological characteristics of the mixed catkins, we selected in six different stages.

We would like to take this opportunity to thank you for all your time involved and this great opportunity for us to improve the manuscript. We hope you will find this revised version satisfactory.

Yours sincerely,

Xuan Zhou

[email protected]

Feng Zou

[email protected]

Reviewer 2 Report

The authors' article is a very extensive and comprehensive study. In my opinion, it is done quite correctly, and the conclusions of the work do not cause doubts.

However, I would advise the authors to make some improvements.

First, the HPLS methodology should be clarified, even in reference 25 there is no mention of the standards used, it is necessary to give the manufacturer's firm and catalog numbers of pure substances or mixtures of standard samples.

Secondly, bars are missing on all histologic photos, it is necessary to correct it.

Third, it is necessary to specify what method of statistical data processing the authors used, what was the post-hoc and p?

Author Response

Dear Reviewer,

Thank you very much for your time involved in reviewing the manuscript and your very encouraging comments on the merits. We also appreciate your clear and detailed feedback and hope that the explanation has fully addressed all of your concerns. In the remainder of this letter, we discuss each of your comments individually along with our corresponding responses. To facilitate this discussion, we first repeat your comments and then present our responses to the comments.

Comments1: The HPLS methodology should be clarified, even in reference 25 there is no mention of the standards used, it is necessary to give the manufacturer's firm and catalog numbers of pure substances or mixtures of standard samples.

Response1: Thank you for making this valuable suggestion. We had replaced reference 25 with a relevant reference and wrote the manufacturer and standard samples as required: The mixed catkins from stage B0-B1, the female flowers from F1-F4, and the male flowers from M1-M4 were preserved at −80 ℃ liquid nitrogen. These frozen samples were ground to fine powder. The levels of endogenous hormones ABA, IAA, GA3, jasmonic acid (JA), zeatin (ZT), brassinosteroid (BR) were extracted with 80% (v/v) methanol and detected by high performance liquid chromatography [25]. Eluted fractions were evaporated, reconstituted with 1 mL 20% (v/v) methanol and injected into a liquid chromatography–electrospray ionization–tandem mass spectrometry apparatus (6410; Agilent, Santa Clara, CA, USA). The standard mother liquor was diluted to 0.1,10,25,50.100 ng/mL standard solution and passed through 0.45 μm organic filter membrane to draw a standard curve. All measurements were conducted in three biological replicates.

Comments2: Bars are missing on all histologic photos, it is necessary to correct it.

Response2: Thank you for pointing this out, bars had been added as required in Figure 2,3.

Comments3: It is necessary to specify what method of statistical data processing the authors used, what was the post-hoc and p?

Response3: Thank you for pointing this out, the original text had been revised as: Microsoft Excel 2022 was used to calculate the raw data of hormone and RT-qPCR. SPSS 24.0 was performed for one-way ANOVA included post hoc test (Duncan test, p<0.05).

We would like to take this opportunity to thank you for all your time involved and this great opportunity for us to improve the manuscript. We hope you will find this revised version satisfactory.

Yours sincerely,

Xuan Zhou

[email protected]

Feng Zou

[email protected]

Reviewer 3 Report

Dear Authors

You have written very well manuscript and covered all the areas of research for this study nicely. In my opinion it should be some minor corrections like:

Keywords should be modified and different from the title.

Follow the same format in whole manuscript for pattern to write headings.

How many replications you used in this study? Please mention how many replications biological or technical for this study.

As you mentioned that correlation for female flowers to the hormone then did you not get any relation in male flowers too?? As you mentioned in your abstract.

Figures should be high quality as there are some figures are low quality.

Author Response

Dear Reviewer,

Thank you very much for your time involved in reviewing the manuscript and your very encouraging comments on the merits. We also appreciate your clear and detailed feedback and hope that the explanation has fully addressed all of your concerns. In the remainder of this letter, we discuss each of your comments individually along with our corresponding responses. To facilitate this discussion, we first repeat your comments and then present our responses to the comments.

Comments1: Keywords should be modified and different from the title.

Response1: Thank you. We changed keywords into: Castanea mollissima; floral morphology; phytohormone; RAN-seq; flower differentiation.

Comments2: Follow the same format in whole manuscript for pattern to write headings.

Response2: Thank you. We checked and revised it.

Comments3: How many replications you used in this study? Please mention how many replications biological or technical for this study.

Response3: All the experiments were conducted in three biological replicates, specific details could be found in the manuscript.

Comments4: As you mentioned that correlation for female flowers to the hormone then did you not get any relation in male flowers too?? As you mentioned in your abstract.

Response4: There were too many male flowers in chestnut, so we were more concerned about how to regulate the male flowers into female flowers and increased the number of female flowers, in order to improve the yield of chestnut. Next, we will investigate which hormones were related to male flowers in future.

Comments5: Figures should be high quality as there are some figures are low quality.

Response5: Thank you for pointing this out, the quality of figures had been improved to 400dpi. You could see the new figures in attachment.

We would like to take this opportunity to thank you for all your time involved and this great opportunity for us to improve the manuscript. We hope you will find this revised version satisfactory.

Yours sincerely,

Xuan Zhou

[email protected]

Feng Zou

[email protected]

Reviewer 4 Report

Dear authors, although the document is well organized and written, I have some concerns about the data presentation and the  results section  (exogenous applications of hormones):

1) Main concerns: 

4.4. Exogenous ZT Treatment Alters the Numbers of Female Flower.  This section of the work placed in this part of manuscript doesn't fit with the rest of the results. If the authors  had evaluated the expression profiles of some key genes after exogenous hormone applications, this telling storyline would make more sense. As presented in the current manuscript, it looks like an independent story.

Suggestion: ¿What do you think about moving section 4.4 after 3.2?

- What criteria did you use to choose the TF IDD7 amongst all the genes found in the pink module (table 3)? There is another TF in that table (BBX19). Why not BBx19?

-Check the figure resolution, please. In almost all figures, the resolution and size are too small, so it is impossible to read them. V.g.: Fig 4. B-D, Fig. 6C-E.

- Figure 1 and 2: Scale bars are missing.

Minor concerns:

-L94: It is more common to use N for North latitude and E for East longitude.

-It isn’t clear what stainings were used in Figures 1 and 2 (H&E or Safranine O -fast green). 

- Endogenous control gene: ACT. ACT2? Which ACT gene was used? Provide reference of ACT gene as control for qRT-PCR in Castanea, please.

-Table 2 format: Homogenize the format of the column title (some are written using Title case or others using lowercase).

-L263-267: Homogenize the module name format: Some names are written using Capital letter and another not.

Minor editing of English language required.

Author Response

Dear Reviewer,

Thank you very much for your time involved in reviewing the manuscript and your very encouraging comments on the merits. We also appreciate your clear and detailed feedback and hope that the explanation has fully addressed all of your concerns. In the remainder of this letter, we discuss each of your comments individually along with our corresponding responses. To facilitate this discussion, we first repeat your comments and then present our responses to the comments.

Comments1: 4.4. Exogenous ZT Treatment Alters the Numbers of Female Flower. This section of the work placed in this part of manuscript doesn't fit with the rest of the results. If the authors had evaluated the expression profiles of some key genes after exogenous hormone applications, this telling storyline would make more sense. As presented in the current manuscript, it looks like an independent story.

Suggestion: ¿What do you think about moving section 4.4 after 3.2?

Response1: Thank you for this very insightful comment. We put 2.7 after 2.3, 3.11 after 3.2, 4.4 after 4.2. The part of expression profiles of some key genes was put after the part of exogenous hormone applications to make the whole story more harmonious.

Comments2: What criteria did you use to choose the TF IDD7 amongst all the genes found in the pink module (table 3)? There is another TF in that table (BBX19). Why not BBx19?

Response2: In the core gene co-expression network, the redder the gene module color is, the more crucial it is. Besides, BBX19 functioned as a novel regulator of the circadian clock was related to the regulation of flowering time, IDD7 functioned as a flowering regulatory transcription factors, associated to flower development. Thus, we selected IDD7 rather than BBX19.

Comments3: Check the figure resolution, please. In almost all figures, the resolution and size are too small, so it is impossible to read them. V.g.: Fig 4. B-D, Fig. 6C-E.

Response3: Thank you for pointing this out, the quality of figures had been improved to 400dpi. You could see the new figures in attachment.

Comments4: Figure 1 and 2: Scale bars are missing.

Response4: Thank you for pointing this out, bars had been added as required in Figure 2,3.

Comments5: L94: It is more common to use N for North latitude and E for East longitude

Response5: North latitude and East longitude had been abbreviated as required.

Comments6: It isn’t clear what stainings were used in Figures 1 and 2 (H&E or Safranine O -fast green).

Response6: Figure 2C,D and Figure 3I are stained by Safranine O -fast green, others in Figures 2 and 3 are stained by H&E. We added a description in the Figures 2 and 3’s annotation.

Comments7: Endogenous control gene: ACT. ACT2? Which ACT gene was used? Provide reference of ACT gene as control for qRT-PCR in Castanea, please.

Response7: We used ACTIN as the control gene, the reference of ACTIN gene as control for RT-qPCR in Castanea was added.

Comments8: Table 2 format: Homogenize the format of the column title (some are written using Title case or others using lowercase).

Response8: Thank you for pointing this out, the format of the column title had been homogenized.

Comments9: L263-267: Homogenize the module name format: Some names are written using Capital letter and another not.

Response9: Thank you for pointing this out, the module name format had been homogenized.

We would like to take this opportunity to thank you for all your time involved and this great opportunity for us to improve the manuscript. We hope you will find this revised version satisfactory.

Yours sincerely,

Xuan Zhou

[email protected]

Feng Zou

[email protected]

Reviewer 5 Report

The manuscript, "Transcription Factor and Zeatin Co-Regulate Mixed Catkin Differentiation of Chinese Chestnut (Castanea mollissima)", by Zhou et al., addressed the morphology of male and female floral organs, underlying mechanisms, gene expression related to biosynthesis and signalling pathways, and transcription factors. They also investigated the effects of jasmonic acid (JA) and zeatin (ZT) exogenous hormone application on flower development in C. mollissima' Tanqiao'n. The manuscript is novel and intriguing. The experimental and statistical approaches are sound, and the literature review strongly supports them. The results are well presented, and the latest references appropriately interpret the data. However, the following question should be addressed for more clarification:

1. Auxins and cytokinesis are responsible for femaleness. However, the authors focused solely on investigating cytokinins due to their specific interest in understanding their role in femaleness. As auxins are known to play an essential role in this process, please clarify. 

2. Line 368: Yan et al. citation is missing.

3. Make sure to highlight future scope in the conclusion part since it will strengthen knowledge on this topic.

Author Response

Dear Reviewer,

Thank you very much for your time involved in reviewing the manuscript and your very encouraging comments on the merits. We also appreciate your clear and detailed feedback and hope that the explanation has fully addressed all of your concerns. In the remainder of this letter, we discuss each of your comments individually along with our corresponding responses. To facilitate this discussion, we first repeat your comments and then present our responses to the comments.

Comments1: Auxins and cytokinesis are responsible for femaleness. However, the authors focused solely on investigating cytokinins due to their specific interest in understanding their role in femaleness. As auxins are known to play an essential role in this process, please clarify.

Response1: We added two references (40,41) in 4.2 to clarify the effect of auxins during flower differentiation.

Comments2: Line 368: Yan et al. citation is missing.

Response2: Thank you for pointing this out, this citation had been added.

Comments3: Make sure to highlight future scope in the conclusion part since it will strengthen knowledge on this topic.

Response3: We agree with this comment. The future scope was added in the conclusion part: But whether LOG interacted with IDD7 remains to be further studied.

We would like to take this opportunity to thank you for all your time involved and this great opportunity for us to improve the manuscript. We hope you will find this revised version satisfactory.

Yours sincerely,

Xuan Zhou

[email protected]

Feng Zou

[email protected]

Reviewer 6 Report

The manuscript: "Transcription Factor And Zeatin Co-Regulate Mixed Catkin Differentiation of Chinese Chestnut (Castanea mollissima)" is well-written and interesting. The authors' objectives were to examine the morphology of male and female floral organs, measure endogenous hormone levels in flowers, profile gene expression related to biosynthesis and signalling pathways and transcription factors, and investigate the effects of jasmonic acid and zeatin exogenous hormones application on flower development in C. mollissima. The paper addresses important issues, uses adequate methods and applies appropriate analyses. It is well structured, and the discussion reflects the results well. The main results provide essential information for understanding the mechanism of mixed catkins development in Chinese chestnut.

Some remarks:

Line 383: CEPA - Please show what the acronym means.
Line 397: CPPU - Please show what the acronym means.
Line 136: qRT-PCR - Everywhere in the text must be replaced with RT-qPCR.
In the Materials and Methods section, 2.6.: Write the internal control used.
Table 1: Write the direction of the primer sequences.
Line 308: 3.9. "DEGs associated with zeatin and jasmonic acid." In this formulation, this sentence is incomplete and unclear.
Line 309: "According to the results of our previous work..." Is this work published or not?
Table 5: What does ± mean? What does the letter after this symbol mean? It is necessary to write!
Line 368: Yan et al. Year, Reference...?

Author Response

Dear Reviewer,

Thank you very much for your time involved in reviewing the manuscript and your very encouraging comments on the merits. We also appreciate your clear and detailed feedback and hope that the explanation has fully addressed all of your concerns. In the remainder of this letter, we discuss each of your comments individually along with our corresponding responses. To facilitate this discussion, we first repeat your comments and then present our responses to the comments.

Comments1: Line 383: CEPA - Please show what the acronym means.
Response1: Thank you for pointing this out, CEPA is ethephon, It revised in the manuscript.

Comments2: Line 397: CPPU - Please show what the acronym means.
Response2: Thank you for pointing this out, CPPU is forchlorfenuron, It revised in the manuscript.

Comments3: Line 136: qRT-PCR - Everywhere in the text must be replaced with RT-qPCR.
Response3: qRT-PCR - Everywhere in the text had been replaced with RT-qPCR.

Comments4: In the Materials and Methods section, 2.6.: Write the internal control used.
Response4: ACTIN was used as the reference gene, we had added it in 2.6.

Comments5: Table 1: Write the direction of the primer sequences.
Response5: We had written the direction of the primer sequences, you can see it in Table 1.

Comments6: Line 308: 3.9. "DEGs associated with zeatin and jasmonic acid." In this formulation, this sentence is incomplete and unclear.
Response6: We revised the title 3.9. as ‘DEGs Related to Zeatin and Jasmonic Acid During Flower Differentiation’.

Comments7: Line 309: "According to the results of our previous work..." Is this work published or not?
Response7: Thank you for pointing this out, we revised this sentence as ‘According to the results of KEGG enrichment analysis and time series analysis’.

Comments8: Table 5: What does ± mean? What does the letter after this symbol mean? It is necessary to write!

Response8: According to one-way ANOVA analysis, data are expressed as mean ± standard deviation, ± means plus or minus ; Different letters in the same column indicate significant difference at 0.05 level, while the same letter indicates no significant difference.

Comments9: Line 368: Yan et al. Year, Reference...?

Response9: Thank you for pointing this out, this reference had been added.

We would like to take this opportunity to thank you for all your time involved and this great opportunity for us to improve the manuscript. We hope you will find this revised version satisfactory.

Yours sincerely,

Xuan Zhou

[email protected]

Feng Zou

[email protected]

Round 2

Reviewer 4 Report

I have read the new version of the manuscript. At this point I don't have more comments.